# Dystrophin-Glycoprotein Complex Behavior in Sternocleidomastoid Muscle of High- and Low-Ranking Baboons: A Possible Phylogenetic Arrangement

**DOI:** 10.3390/jfmk7030062

**Published:** 2022-08-25

**Authors:** Antonio Centofanti, Giovanna Vermiglio, Giuseppina Cutroneo, Angelo Favaloro, Giacomo Picciolo, Felice Festa, Giuseppe Pio Anastasi

**Affiliations:** 1Department of Biomedical and Dental Sciences and Morphofunctional Imaging, University of Messina, 98124 Messina, Italy; 2Department of Innovative Technologies in Medicine and Dentistry, University “G. D’Annunzio” of Chieti-Pescara, 66100 Chieti, Italy

**Keywords:** baboon, sternocleidomastoid muscle, immunofluorescence, SGC, DGC, dystroglycans, dystrophin

## Abstract

The dystrophin-glycoprotein complex is a multimeric system made up of the sarcoglycan sub-complex, the sarcomplasmatic complex and the dystroglycans complex. The sarcoglycan sub-complex stabilizes the sarcolemma during muscle activity and plays a role in force transduction. This protein system is also expressed in the muscle of non-human primates such as chimpanzees and baboons, and its expression changes depending on social ranking. In fact, previous data have shown that all muscle fibers of masseter and sternocleidomastoid muscles of chimpanzees and high- ranking baboons always express sarcoglycans, while middle- and low-ranking baboons are characterized by fibers that are negative for the sarcoglycan sub-complex. Given this information, the aim of the present work was to evaluate the expression of other proteins such as laminin, beta dystroglycan and dystrophin in the sternocleidomastoid muscle of high- and low-ranking baboons. The samples were processed by immunohistochemistry; results show that in high-ranking baboons, all tested proteins were always expressed while in low-ranking baboons, fibers that were negative for sarcoglycans and beta dystroglycan have been observed. No negative fibers for laminin and dystrophin have been found in low-ranking baboons suggesting that only the transmembrane proteins of the dystrophin glycoprotein complex change in their expression and that could be correlated to a phylogenetic arrangement.

## 1. Introduction

The Dystrophin-Glycoprotein Complex (DGC) is a multimeric system made up of the sarcoglycan sub-complex (SGC), the sarcomplasmatic complex and the dystroglycan complex. The sarcoglycan sub-complex (SGC) is a protein transmembrane system made up of six isoforms: alpha, beta, gamma, delta, epsilon and zeta-sarcoglycans [1]. The sarcoplasmatic complex is made up of dystrophin, dystrobrevin and syntrophins; the dystroglycan complex is formed by alpha and beta-dystroglycans [2,3,4].

The SGC, together with the dystroglycan complex, represent the transmembrane proteins that directly and indirectly link other protein systems inside and outside muscle fibers. In detail, the SGC links the beta-dystroglycan (beta-DG) by a lateral link that in turn links to the alpha-dystroglycans. The alpha-dystroglycan links the laminin at an extracellular level. Intracellularly, the beta dystroglycan links the dystrophin [5].

The DGC plays a key role in connecting the cytoskeleton to the extracellular matrix. By this linkage, they stabilize the sarcolemma during muscle activity and play a role in force transduction. The importance of DGC is evidenced by several pathologies that arise from the mutation of dystrophin and some sarcoglycans (SGs) isoforms—these pathologies are called dystrophies [6,7,8]. In particular, the absence of SGs determines the sarcoglycanopathy, and this represent the unique condition in which muscle fibers are negative for SGs [9,10,11,12].

The reduction of SGs have been observed in other muscle pathologies or in pathology that involve muscle function [13,14]. Several studies have shown the reduction in SGs expression in masticatory muscles during crossbite [15,16,17,18,19]. Our results have shown that these proteins decrease in muscle affected by crossbite and increase in contralateral hypertrophic muscle. This suggests that these proteins are tightly connected to muscle strength: their expression seem to increase or decrease when muscle strength increases or decreases, respectively.

The DGC is a complex that is expressed not only in human but also in muscles of chimpanzees that share 99% of their genetic code with humans [20,21,22,23].

Additionally, in primates, the expression of SGC seem to be linked to muscle strength. In fact, in masticatory muscle of chimpanzees we found an increased expression of SGs in alpha male chimpanzees when compared to non-alpha males [24]. In fact, the masseter muscle of alpha males is characterized by a higher contraction force that is typical of primates that should defend their territory and other members of the group [24].

Intriguing results have been obtained about SGC expression in baboons; these primates are phylogenetically far from humans when compared to chimpanzees and share 96% of their genetic code with humans [25]. The social organization of baboons is made up of three rank classes: (1) high, middle and low. It has been found that in high-ranking baboons, all masseter and sternocleidomastoid fibers were positive for SGC; in middle and low dominance classes, 15% and 50% of fibers, respectively, were negative for SGC. The absence of SGC in 50% of fibers of healthy, low-ranking baboons allows us to hypothesize the existence of an alternative protein system [26]. Our hypothesis is that other protein systems related to the SGC could change in their levels of expression depending on a baboon’s ranking.

The aim of the present work is to verify if DGC members change in their expression between the different class rankings within primates. For these reasons, we investigated the expression of alpha, beta, gamma, delta, epsilon and zeta-sarcoglycans and the expression of beta-dystroglycan, laminin and dystrophin in the sternocleidomastoid muscle of baboons of high- and low-rankings.

## 2. Materials and Methods

### 2.1. Muscle Biopsies

In the present study, we examined biopsies of tissue fragments of sternocleidomastoid muscles of baboons belonging to different social ranks. Muscle biopsies were performed after treating the animals under anesthesia. After a skin incision was made to expose the muscle, specimens were collected with a 6.0 mm biopsy punch. The biopsies are kept in the phylogeny section of The Ancient Anatomical Museum of the University of Messina.

### 2.2. Immunofluorescence Reactions

Muscle biopsies were fixed in 4% paraformaldehyde in a 0.2 M phosphate buffer with a pH of 7.4 for 2 h at room temperature. After numerous rinses in a 0.2 M phosphate buffer with a pH of 7.4 and phosphate-buffered Saline (PBS), the biopsies were infiltrated with 12–18% sucrose and then snap-frozen in liquid nitrogen. Some 20 m thick sections were cut with a cryostat and placed on glass slides coated with 0.5% gelatin and 0.005% chromium potassium sulfate. In order to block nonspecific binding sites and permeabilize membranes, the sections were pre-incubated with 1% bovine serum albumin and 0.3% Triton X-100 in PBS for 30 min at room temperature. Finally, the sections were incubated with primary antibodies. The following primary polyclonal antibodies were used with a 1:100 dilution: anti-α-SG, anti-β-SG, anti-γ-SG, anti-δ-SG, anti-ε-SG and anti-ζ-SG (Santa Cruz Biotechnology, Santa Cruz, CA, USA). To perform the double reaction between sarcoglycans we used two SGs with different source, rabbit and goat. We also use anti-beta-dystroglycan antibody (SigmaAldrich, St. Louis, MI, USA), diluted 1:250, anti-Laminin antibody (SigmaAldrich, St. Louis, MI, USA) diluted 1:200 and anti-dystrophin antibody (SigmaAldrich, St. Louis, MI, USA) diluted 1:300. Primary antibodies anti SGs were demonstrated using Fluorescein isothiocyanate (FITC)-conjugated IgG (Jackson ImmunoResearch Laboratories, Inc., West Grove, PA, USA) while beta-dystroglycan, laminin and dystrophin antibodies were detected using Texas Red-conjugated IgG (Jackson ImmunoResearch Laboratories, Inc., West Grove, PA, USA). Sections were then observed, and images were captured with a Zeiss LSM 5 DUO (Carl Zeiss) confocal laser scanning microscope. All images were digitalized with an 8-bit resolution into an array of 2048 × 2048 pixels. Optical sections of fluorescent specimens were obtained using an HeNe laser (543 nm) and an argon laser (458 nm) at a 1-min, 2-sec scanning speed with up to eight averages; 1.50-µm-thick sections of fluorescent specimens were obtained using a pinhole of 250. Contrast and brightness were established by examining the most brightly labeled pixels and choosing settings that allowed clear visualization of the structural details while keeping the highest pixel intensities close to 200. The same settings were used for all images obtained from the other samples that had been processed in parallel. For image analysis, we used a function called “splitting” to show individual channels and relative merge. For each reaction, at least 100 individual fibers were examined. The Z-stack function has been used to perform a 3D reconstruction of fluorescence across the entire section’s thickness. Digital images were cropped, and figure montages were prepared using Adobe Photoshop 7.0 (Adobe System, Palo Alto, CA, USA).

## 3. Results

### 3.1. Sarcoglycans Immunoreactions

In the present work, we performed double immunofluorescence reactions for SGs with the following combinations: alpha-beta; alpha-gamma; alpha-delta; alpha-epsilon; alpha-zeta; beta-gamma; beta-delta; beta-epsilon; beta-zeta; gamma-delta; gamma-epsilon; gamma-zeta; delta-epsilon; delta-zeta; epsilon-zeta. Results show that the staining pattern for all SGs is uniformly distributed along the sternocleidomastoid muscle of high-ranking baboons and all the observed fibers were positive for SGs (Figure 1). Results obtained in the sternocleidomastoid muscle of low-ranking baboons show the presence of negative and positive fibers for alpha- and beta-SGs (Figure 2A–D) and for gamma- and delta SGs (Figure 2E–H) in the same microscopic fields, as evidenced. The negative fibers are evidenced by with asterisks and arrows (Figure 2). The double immunoreaction showed that all the SGs isoforms move together: in fact, the negative fibers are negative for all the isoforms and the positive fibers are positive for all the SGs. The 3-dimensional reconstruction function of confocal laser microscope “z-stack” was used to observe immunostaining across the entire thickness of the sections (16 µm) obtaining a gallery with 21 pictures, each with a scan step size of 1 µm in a double localization reaction. This gallery shows that in the sternocleidomastoid muscle of low-ranking baboons, the SG staining pattern in the positive fibers change depending on the focus, while in the negative fibers, the SG staining pattern is always undetectable. This confirms that the fibers without an SG staining pattern really are negative for these proteins and that such a pattern does not depend on microscope focus (Figure 3).

### 3.2. Sarcoglycans/Beta-DG Immunoreactions

We also performed double immunofluorescence reactions between each SG and beta-dystroglycan. Results show that the staining pattern for all SGs and beta-dystroglycan is uniformly distributed along the sternocleidomastoid muscle of high-ranking baboons, and all the observed fibers were positive for SGs and beta-dystroglycan (Figure 4). Results obtained in the sternocleidomastoid of low-ranking baboons shows the presence of negative and positive fibers for both alpha-SGs/beta-dystroglycan (Figure 5A–D) and gamma-SGs/beta-dystroglycan (Figure 5E–H) in the same microscopic fields. Negative fibers are evidenced by white asterisks and arrows (Figure 5). The double immunoreaction showed that both SGs and beta-dystroglycan move together: in fact, the negative fibers are negative for all SGs and beta-DG, and the positive fibers are positive for all the SGs and beta-DG. The same data have been found for all tested sarcoglycans. The 3-dimensional reconstruction function of confocal laser microscope “z-stack” was used to observe the immunostaining across the entire thickness of the sections (16 µm) obtaining a gallery with 21 pictures, each with a scan step size of 1 µm in a double localization reaction. This gallery shows that in the sternocleidomastoid muscle of low-ranking baboons, the SG/beta-DG staining pattern in the positive fibers changes depending on the focus, while in the negative fibers, the SG/beta-DG staining pattern is always undetectable. This confirms that the fibers without SG/beta-DG staining pattern are truly negative for these proteins and that this does not depend on the focus (Figure 6).

### 3.3. Laminin Immunoreaction

Results of laminin immunofluorescence show the presence of a staining pattern along the contour of fibers at the basal lamina level in the sternocleidomastoid of both high- and low-ranking baboons. This time, no fibers negative for laminin have been observed in low-ranking baboons. Moreover, no difference in the pattern of fluorescence for laminin between samples of high-ranking and low-ranking baboons has been observed (Figure 7).

### 3.4. Sarcoglycans/Dystrophin Immunoreactions

Double immunofluorescence reactions between each SGs and dystrophin show that the staining pattern for all SGs and dystrophin is uniformly distributed along sternocleidomastoid muscle of high-ranking baboon and all the observed fibers were positive for alpha-SGs and dystrophin (Figure 8A–C). Results obtained in sternocleidomastoid of low-ranking baboons show the presence of negative and positive fibers for alpha-SGs in the same microscopic fields (Figure 8D, white arrow) while the fibers were always positive for dystrophin (Figure 8F, white arrow). The double immunoreaction shows that SGs and dystrophin do not move together.

## 4. Discussion

In the present study, we analyzed the expression of several protein systems that belong to the dystrophin glycoprotein complex (DGC) in sternocleidomastoid muscles from healthy high- and low-ranking baboons. In detail, we tested the expression of all SGCs and also of beta-dystroglycan, laminin and dystrophin. Our results show that in high-ranking baboons, all muscle fibers are positive for all tested proteins. On the other hand, in low-ranking baboons, fibers were negative for all sarcoglycans isoforms of beta-dystroglycans, but no fibers that were negative for laminin and dystrophin have been found. Moreover, sarcoglycans and beta dystroglycan were always found lacking in the same fibers, showing that these two protein systems seem to move together.

In the past, we have already observed the expression of the SGC in the masseter and sternocleidomastoid of baboons of high-, middle- and low-ranking with original and intriguing results: 50% of fibers of low-ranking baboons were negative for all sarcoglycan isoforms; middle-ranking baboons have 15% of fibers negative for sarcoglycans; while the high-ranking baboons have fibers that were only positive for all sarcoglycans [26]. It is well known that the SGC and the DGC are dynamic systems that modulate their shape depending on the stimuli to which they are subjected during myogenesis and adult muscle function [27]. They are especially involved in mechanical force transduction and sarcolemma stabilization, playing a key role in muscle maintenance and adaptation to muscle strength [28,29]. We have already demonstrated the role of SGC in the restoration of muscle strength in the human masseter muscle of subjects with unilateral cross-bite: in the affected side, characterized by a low muscle strength, the expression of SGC decreases when compared to the contralateral side where the muscle strength increases. Given this, we hypothesized that SGC expression is correlated to high muscle strength [15,16].

Our findings, showing healthy fibers of low-ranking baboon negative for SGC and beta-DG, strongly support the role of DGC in contractile force maintenance. As Cutroneo et al. (2015) have already explained, the different staining patterns of these proteins could be correlated to the social behavior of baboons. Human and non-human primates socially interact following two different social models: agonic and hedonic. The agonic model, characteristic of Rhesus macaques and baboons, is based on threatening behaviors and high stress, typical of competitive systems. The hedonic model is based on social behaviors aimed at collaboration, in which the well-being of the group prevails over the individual. Humans are characterized by both social models [30,31]. Male baboons constantly compete for leadership that comes from demonstrating strength and superiority. When the baboons reach a high rank, they are characterized by energy-intensive mate-guarding events with high muscle strength involvement [32,33,34,35]. Consequently, these proteins, which play a key role in muscle activity by controlling contractile forces, may influence the phenotype of muscle fibers that determines the adaptation to new functional conditions.

The present work shows that not only does the SGC change in its expression between the different ranking classes, but beta-dystroglycan changes also. This protein, together with alpha-dystroglycan, forms an important protein system that mediates interaction between the cytoskeleton and the extracellular matrix, contributing to sarcolemma stabilization and force transduction. The interactors such as laminin and dystrophin are always expressed in all class rankings.

By this, it is possible to hypothesize that the muscles of high-ranking baboons would need 100% of positive fibers for these proteins systems in order to adapt muscle strength to the motor tasks typical of this social class.

This suggest that epigenetic modification of the muscular protein network would be responsible for phylogenetic variation. It is possible to assess that this variation involves only those protein systems of the DGC that are located at the plasma membrane level, while the intracellular and extracellular interactors are always present. Therefore, several negative muscle fibers of low-ranking baboons could be characterized by the expression of ancestral transmembrane proteins that cannot yet be detected by the usual antibodies. The strength of this work is the analysis of an important protein system in baboon muscle that is a difficult sample to obtain today. Likewise, the limitation of the work is that due to the difficulty of obtaining further baboon biopsies, this work is difficult to reproduce.

## 5. Conclusions

The absence of fibers negative for DGC in chimpanzees and the existence of negative fibers for SGs and beta-dystroglycan in low-ranking baboons suggest that the protein systems expression could be correlated to phylogenetic mechanisms. In fact, in the evolutionary scale of primates, chimpanzees are very close to human when compared to baboons. This strongly suggest that these proteins’ variation could be linked to epigenetic pathways on the basis of the evolution.

## Figures and Tables

**Figure 1 jfmk-07-00062-f001:**
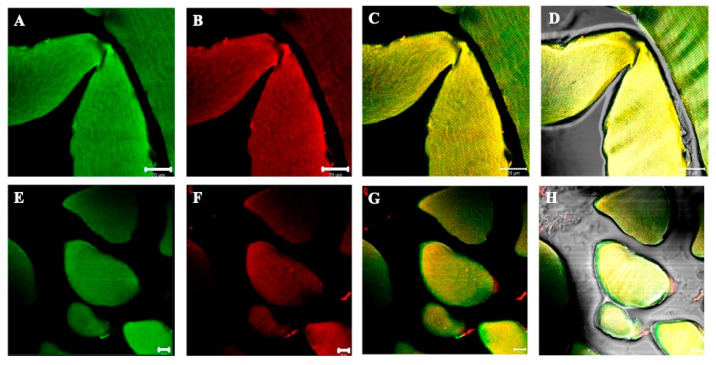
Compound panel of double immunofluorescence reaction between alpha-SG (green channel) and beta-SG (red channel) (**A**–**D**) and between gamma- and delta-SG (**E**–**H**) performed on sternocleidomastoid of high-ranking baboons. Results show that both alpha- and beta-SG are expressed along all observed muscle fibers. The transmitted light shows that all the fibers in the microscopic field are positive for sarcoglycans (**D**,**H**). The yellow fluorescence arises from the merge between green and red fluorescence and corresponds to protein colocalization. The same data have been found for all the SG isoforms.

**Figure 2 jfmk-07-00062-f002:**
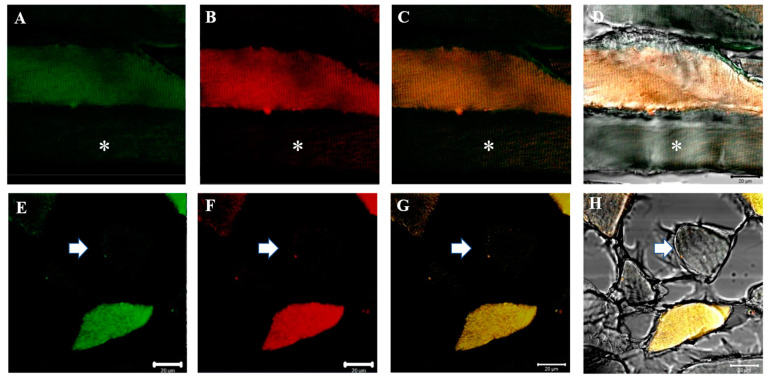
Compound panel of double immunofluorescence reaction between alpha-SG (green channel) and beta-SG (red channel) (**A**–**D**) and between gamma-SG and delta-SG (**E**–**H**) performed on sternocleidomastoid of low-ranking baboons. Results show the presence of negative fibers for both alpha- and beta-SGs (**A**–**D**, asterisk) and for both gamma- and delta-SGs (**E**–**H**, arrow). The transmitted light shows the negative fibers (**D**,**H**). Positive and negative fibers are in the same microscopic field. The yellow fluorescence arises from the merge between green and red fluorescence. The same data have been found for all the SG isoforms.

**Figure 3 jfmk-07-00062-f003:**
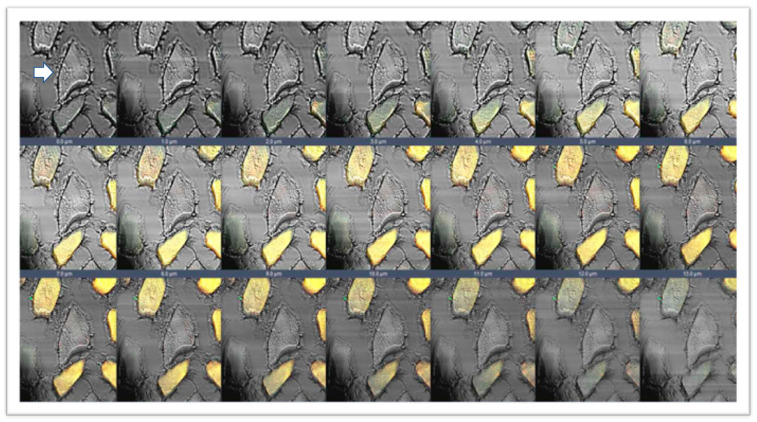
Gallery with 21 pictures of 1 μm thickness each obtained by z-stack function of confocal laser microscope. The figure shows a double immunofluorescence reaction between alpha- and beta-SGs (yellow fluorescence) in the sternocleidomastoid muscle of low-ranking baboons. This 3D reconstruction confirms the presence of negative fibers for both alpha and beta-SGs along the thickness of the section, as evidenced by transmitted light (arrow). The same data have been found for all the SG isoforms.

**Figure 4 jfmk-07-00062-f004:**
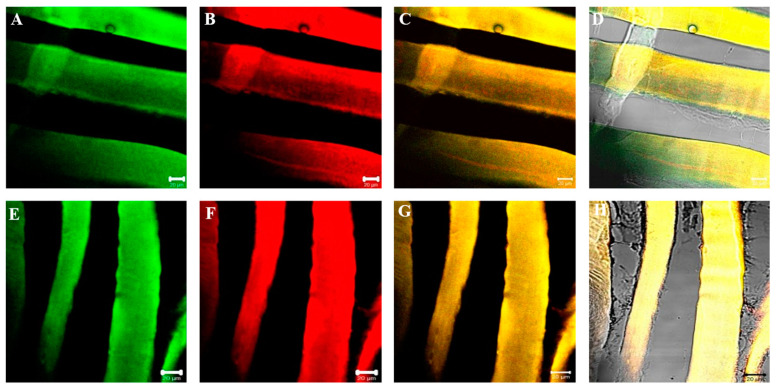
Compound panel of double immunofluorescence reaction between alpha-SG (green channel) and beta-dystroglycan (red channel) (**A**–**D**) and between gamma-SG and beta-dystroglycan (**E**–**H**) performed on sternocleidomastoid of high-ranking baboons. Results show that both alpha- and gamma-SGs and beta-dystroglycan are expressed along all observed muscle fibers. The yellow fluorescence arises from the merge between green and red fluorescence. The transmitted light shows that all the fibers in the microscopic field are positive for the tested proteins (**D**,**H**). The same data have been found for all the SG isoforms.

**Figure 5 jfmk-07-00062-f005:**
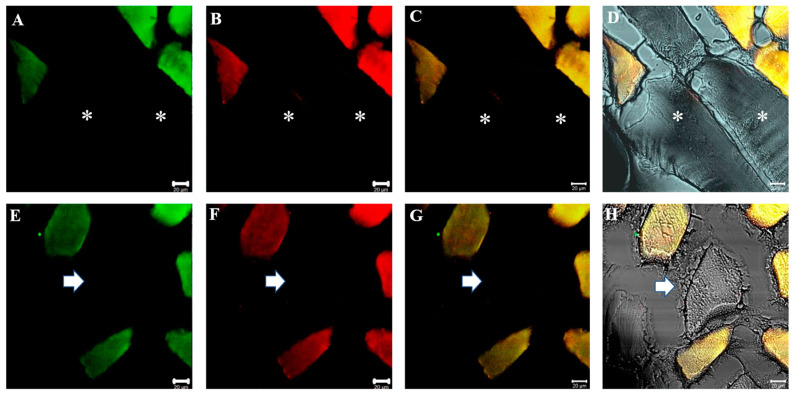
Compound panel of double immunofluorescence reaction between alpha-SG (green channel) and beta-dystroglycan (red channel) (**A**–**D**) and between gamma-SG and beta-dystroglycan (**E**–**H**) performed on sternocleidomastoid of low-ranking baboons. Results show the presence of negative fibers for both alpha-SG and beta dystroglycan (**A**–**D**, asterisks) and for both gamma-SG and beta-dystroglycan (**E**–**H**, arrow). Positive and negative fibers are in the same microscopic field as evidenced by transmitted light (**D**,**H**). The yellow fluorescence arises from the merge between green and red fluorescence. The same data have been found for all the SG isoforms.

**Figure 6 jfmk-07-00062-f006:**
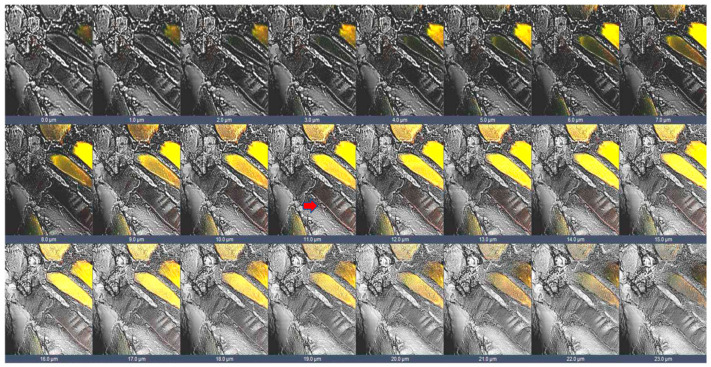
Gallery with 24 pictures of 1 μm thickness each obtained by z-stack function of confocal laser microscope. The figure shows double immunofluorescence reaction between alpha- and beta-dystroglycan (yellow fluorescence) in sternocleidomastoid muscle of low-ranking baboons. This 3D reconstruction confirms the presence of negative fibers for both alpha-SG and beta-dystroglycan along the thickness of section, as evidenced by transmitted light (red arrow). The same data have been found for all the SG isoforms.

**Figure 7 jfmk-07-00062-f007:**
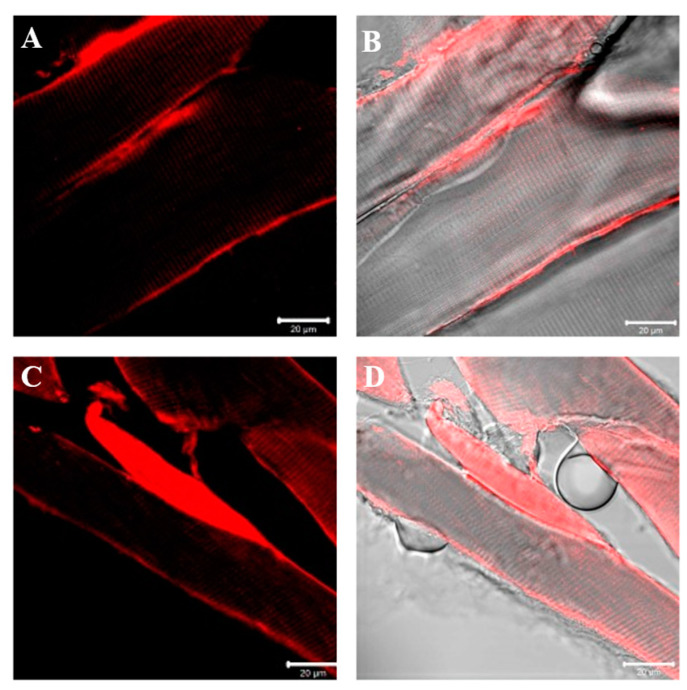
Compound panel of single immunofluorescence reaction using anti-laminin antibody in sternocleidomastoid muscle of high-ranking (**A**,**B**) and low-ranking (**C**,**D**) baboons. Laminin fluorescence pattern (red channel) is distributed around all muscle fibers of both ranking classes, at basal lamina level as evidenced by transmitted light (**B**,**D**).

**Figure 8 jfmk-07-00062-f008:**
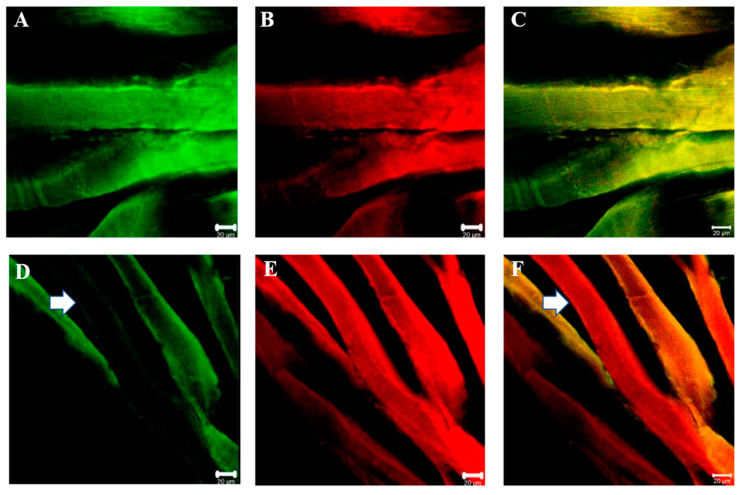
Compound panel of double immunofluorescence reaction between alpha-SG (green channel) and dystrophin (red channel) in sternocleidomastoid muscle of high-ranking (**A**–**C**) and low-ranking (**D**–**F**) baboons. Results show that all muscle fibers of high-ranking baboon are positive for both alpha-SG and dystrophin; instead, in low-ranking baboons, all fibers are positive for dystrophin while there are some negative fibers for alpha-SG (white arrow).

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
