# Peer review of "Dystrophin-Glycoprotein Complex Behavior in Sternocleidomastoid Muscle of High- and Low-Ranking Baboons: A Possible Phylogenetic Arrangement"

_jfmk, 2022, doi:10.3390/jfmk7030062_

Round 1

Reviewer 1 Report

Title:Dystrophin-glycoprotein complex behaviour in sternocleidomastoid muscle of high- and low-ranking baboons: a possible phylogenetic arrangement

This article seems well built and brings evidence of a physiological phenomenon not yet fully understood and that certainly deserves further study.

Some hard points of revision, suggested to the authors to increase the level of the paper, are provided below

I suggest to include the specific reference for each device used / reliability and validity

Reviewer 2 Report

Dear authors,

thank you for this very interesting article.

The results are clear and the message is very well presented. Do you expect similar results concerning animals who have less similarities to human beings?

Reviewer 3 Report

The aim of the present work was to evaluate the expression of other proteins such as laminin, beta dystroglycan and dystrophin in sternocleidomastoid muscle of high- and low-ranking baboons. This is a very great study, which i sugegst a few comments prior the acceptance for publication.

-Introduction: To add the hypothesis of study.

-Methods: To clarify the muscle biopsies. This is a short description and did not show how did was performed.

-Results: The quality of fig 6 is not good.

-Discussion: to add a limitation and strengths of study.
